# Immune response against *Chlamydia trachomatis* via toll-like receptors is negatively regulated by SIGIRR

**Mufadhal Al-Kuhlani[1,2], Graham Lambert[3], Sukumar Pal[4], Luis de la Maza[4], David M. Ojcius[2]***

**1** Life Science Department, Fresno City College, Fresno, CA, United States of America, **2** Department of Biomedical Sciences, Arthur Dugoni School of Dentistry, University of the Pacific, San Francisco, CA, United States of America, **3** College of Osteopathic Medicine, Touro University Nevada, Henderson, NV, United States of America, **4** Department of Pathology and Laboratory Medicine, University of California Irvine, Irvine, CA, United States of America

* dojcius@pacific.edu

**Data Availability Statement:** All relevant data are within the paper.

## Abstract

*Chlamydia trachomatis* is the most common bacterial sexually-transmitted infection and the major cause of preventable blindness worldwide. The asymptomatic nature of many infections along with uncontrolled inflammation leads to irreversible damage in the upper genital tract and the tarsal conjunctivae, with the major complications of infertility and chronic pelvic pain, and blindness, respectively. Inflammation must, therefore, be tightly regulated to avoid an unrestrained immune response. The genetic factors that regulate inflammation through Toll-like receptor (TLR) signaling pathways during *C. trachomatis* infection have not been fully characterized. SIGIRR (also known as IL-1R8 or TIR8) can regulate inflammation in response to various pathogens and diseases. However, nothing is known about its role during *C. trachomatis* infection. Expression of the pro-inflammatory chemokine, IL-8, was measured in epithelial cells infected with *C. trachomatis*. The effect of SIGIRR was determined by depleting SIGIRR or over-expressing SIGIRR in the epithelial cells before infection. Our results indicate that, in the absence of SIGIRR, epithelial cells induce higher levels of the pro-inflammatory chemokine, IL-8, in response to *C. trachomatis* infection. In addition, SIGIRR associates with MyD88 in both infected and uninfected infected cells. Collectively, our data demonstrate that SIGIRR functions as a negative regulator of the immune response to *C. trachomatis* infection. This finding provides insights into the immuno-pathogenesis of *C. trachomatis* that can be used to treat and identify individuals at risk of uncontrolled inflammation during infection.

## Introduction

Single Ig IL-1 receptor (ILR)-related molecule (SIGIRR), also known as Toll interleukin-1 receptor (IL-1R) 8 (TIR8) or IL-1R8, is a member of the Toll/IL-1 receptor (TIR) superfamily that includes the Toll-like receptor (TLR) and interleukin-1 receptor (ILR) subfamilies. Unlike the ILR members that contain three Ig domains, SIGIRR has a single domain in the

**Funding:** This work was supported by a grant from the National Institute of Health (R56 AI078419) (DMO and LdM). The funders had no role in study design, data collection and analysis, decision to publish, or preparation of the manuscript.

**Competing interests:** The authors have declared that no competing interests exist.

extracellular region and a TIR domain in the cytoplasmic side of the cell. IL-36Ra has been proposed as a ligand in the brain, and SIGIRR was shown to be a coreceptor for IL-37-IL1R5/ IL-18Rα, and required for the anti-inflammatory effects of IL-37 [1–3]. Recent studies have shown that SIGIRR deficiency in mice results in kidney graft rejection, enhancement of the development of colitis-associated cancer and intestinal inflammation, and increases in susceptibility to lupus and bacterial/fungal infections [4–9]. The immune responses in SIGIRR-deficient mice are associated with induced expression of proinflammatory agents, elevated infiltration of PMNs and increased tissue damage. In addition SIGIRR plays an essential role in the regulation of Th2, although maybe not Th1, responses [10].

SIGIRR does not associate with NF-κB directly, but it triggers an inhibitory effect through its interaction with other adaptors via TIR domains [11]. Computational structural studies suggest that SIGIRR does not block formation of the MyD88-dependent signalosome, but it inhibits NFκB activation by preventing translocation of the signalosome from the receptor [12]. Two reports show that resistance against *P. aeruginosa* infection of lung and cornea is mediated by SIGIRR [11, 13]. In response to a *P. aeruginosa* infection in mice, the neutralization of SIGIRR protein with anti-SIGIRR antibody resulted, in increased damage to the host cells, higher titer of the bacteria, and increased levels of proinflammatory cytokines. Moreover, the inhibitory effect of SIGIRR was dose-dependent since the stimulation of IL-1R1 and TLR4, but not TLR3, reduced NF-κB activity as the expression level of SIGIRR increased [11]. In addition to TLR4, SIGIRR negatively regulates the signaling pathways of TLR2, TLR5, TLR7, and TLR9 in various tissues [9, 14–16]. The TIR domain, but not the extracellular Ig domain, of SIGIRR is necessary for interfering with recruitment of other adaptors to the TLRs [14]. In addition to its ability to self-dimerize, SIGIRR interacts strongly with IL-1R and other signaling molecules such as MyD88, IRAK and TRAF6 [14, 15, 17]. Studies of immature dendritic cells, which express high levels of SIGIRR, indicate that association of MyD88 with SIGIRR through the TIR domain is constitutive and necessary for the dendritic cells to remain immature [18]. Disruption of this connection, or the decrease of SIGIRR's expression, leads to maturation of the dendritic cells and enhances the immune response [4, 18].

Chlamydiae have a unique biphasic developmental life cycle that are capable of infecting various types of tissues in humans and animals [19]. Even though the species *C. trachomatis* causes genital infections that, if diagnosed early, can be treated with antibiotics, severe acute or chronic asymptomatic infections can lead to long-term complications, including pelvic inflammatory disease, ectopic pregnancy and infertility [20–22]. These complications are mainly linked to severe acute infections and to chronic inflammation triggered by repeated or persistent infections. Therefore, inflammation must be tightly regulated to avoid an uncontrolled immune response. Since TLR signaling pathways play a major role in initiating inflammation, negative regulation can be accomplished at multiple levels in TLR signaling pathways.

Epithelial cells lining tissues of the reproductive tract of humans express many pattern-recognition receptors (PRRs), including members of the Toll-like receptors (TLRs) such as TLR1, TLR2, TLR3, and TLR4 [23]. Since epithelial cells are the main target for *C. trachomatis* infection, they are considered the first responder cells that initiate and sustain the release of proinflammatory cytokines against *C. trachomatis* such as tumor necrosis factor-α (TNF-α), granulocyte macrophage-colony stimulating factor (GM-CSF), interleukin (IL)-1, IL-6 and IL-8 [24–27]. However, except for a report that TRAIL-R1 is a negative regulator of inflammation in humans and murine cells [28], few studies have been conducted to understand whether host cells may exploit some parts of TLR pathways to negatively regulate the proinflammatory activities against *C. trachomatis* to avoid an uncontrolled immune response.

The ability of SIGIRR to interact with various components of TLR signaling pathways in a manner that influences the immune response of epithelial cells during bacterial infection

indicates its potential importance as a negative regulator during chlamydial infection. Here, we investigated the role of SIGIRR in regulating inflammation caused by *C. trachomatis* infection in human epithelial cells and some of the mechanisms by which it negatively regulates the immune response.

## Materials and methods

### Growth of cell lines and bacteria

The human cervical carcinoma cell line, HeLa 229, was obtained from American Type Culture Collection (ATCC). HeLa cells were cultured in a humidified incubator at 37˚C with 5% $CO_2$ in DMEM/F12 medium (Invitrogen) supplemented with 10% heat-inactivated fetal bovine serum (FBS) (Invitrogen), 50 U/ml penicillin, 50 μg/ml streptomycin (Invitrogen) and 10 μg/ml gentamycin. The LGV/L2 strain of *C. trachomatis* was obtained from Roger Rank (University of Arkansas, Little Rock, AR) and grown and harvested in HeLa as previously described [29, 30]. Briefly, $1x10^7$ HeLa cells were plated on 150 mm plate until they reached ~ 60% confluence in antibiotic free media. The monolayer of HeLa cells was infected with frozen stock of LGV-L2 EBs at a multiplicity of infection (MOI) of 2 and incubated at 37˚C. Forty-eight hpi, cells were collected using sterile cell scraper and stored at -80˚C. Collected cells were subjected to a freeze-thaw cycle and vortexed for one minute. Cell suspensions were centrifuged for 15 min at 500*xg* at 4˚C. The supernatant was centrifuged again for 30 min at 30,000 *x* g at 4˚C. The supernatant was discarded, and the pellet was resuspended in 2 ml sucrose/phosphate/glutamate buffer (SPG), aliquoted into appropriate volume and stored at -80˚C until ready for use. The number of bacterial inclusion-forming units (IFU) of *C. trachomatis* was determined by infecting HeLa cell monolayer cultures as described previously [30, 31].

HEK-Blue SEAP reporter cell lines Null1, Null2, and hTLR2 (Invivogen, San Diego, CA) were grown in T75 flasks with DMEM high glucose media (Invitrogen) supplemented with 10% FCS (Invitrogen), 50 U/ml penicillin, 50 μg/ml streptomycin (Invitrogen) and selected with the appropriate antibiotic according to the manufacturer's instructions.

### Generation of transient SIGIRR-deficient cells

Transient reduction of SIGIRR was accomplished by transfecting HeLa cells with gene-specific siRNA. Lipofectamine 2000 reagent was used to transfect HeLa cells following the manufacturer's instructions. Briefly, HeLa cells were plated 24 hrs prior to transfection so the cells would be 70% confluent at the time of transfection. Lipofectamine 2000 mixture was prepared in the appropriate amount of serum-free medium Opti-MEM. Cells were transfected with siRNA non-target control (Cell Signaling), SIGIRR-specific siRNA (NM_021805 oligo #1269132 for Seq A, and 1269134 for Seq B) and then were incubated at 37˚C with 5% $CO_2$ for 24 h. Cells were then detached by TrypLE-Express (Invitrogen), and plated to obtain 80% confluency after 18 h before experimental treatment. The knockdown was confirmed via qPCR, as described below.

### Transfection of Flag-tagged-SIGIRR plasmid

For preparation of Flag-tagged cells, $5x10^5$/well of HEK-Blue hTLR2 cells or $5x10^5$/well of HeLa cells were seeded in 6-well plates and incubated at 37˚C and 5% $CO_2$ until they reached 60% confluency. HEK-TLR2 cells were then transfected with SIGIRR-FLAG empty vector pCDNA3.1+ plasmids (kind gift of Dr. Li Xiaoxia, Cleveland Clinic Foundation), at different concentrations for each plasmid as indicated (0.1 and 1.0 μg), using Lipofectamine 2000 reagent (Invitrogen) following the manufacturer's instructions. HeLa cells were transfected

with 4 μg of MyD88 HA and with either SIGIRR-FLAG plasmids or empty vector pCDNA3.1
+ plasmids. Transfection media was changed 6 hours post-transfection and the cells were re-
plated as needed and left to grow for a total of 48 hrs. Cells were then infected with the LGV/
L2 strain of *C. trachomatis* at an MOI of 1. Twenty-four hpi, cells were collected and used for
either qPCR or co-immunoprecipitation assays.

## RNA isolation, reverse transcription, and qPCR

Total RNA was isolated from cells using the Qiagen RNeasy kit (Qiagen) following the manu-
facturer's instructions. The synthesis of the complementary DNA (cDNA) template was
conducted according to the manufacturer's instructions (TaqMan, Roche). The PCR was con-
ducted using Qiagen Fast Cycling PCR Kit. PCR conditions included denaturation steps at
95˚C for 5 min, followed by 35 cycles at 96˚C for 5 sec, 60˚C for 5 sec and 68˚C for 5 sec
followed by 72˚C for 1 min. The PCR products were separated by 2% agarose gel electrophore-
sis and visualized using ethidium bromide. The following primers were used: GAPDH: for-
ward, 5'-TTAAAAGCAGCCCTGGTGAC-3'; reverse: 5'CTCTGCTCCTCCTGTTCGAC-3'
(144bp); SIGIRR: forward- 5'- TCAGTGGCTCTGAACTGCAC-3', reverse: 5'- GTACCA
GAGCAGCACGTTGA-3' (352bp); IL-8: forward, 5'-AATCTGGCAACCCTAGTCTGCTA-3';
reverse, 5'- AGAAACCAAGGCACAGTGGAA-3' (64 bp). The q-PCR was conducted in trip-
licates of 20 μl final volume using Mx3000P (Stratagene, La Jolla, CA) with Brilliant III Ultra
Fast STBR Green qPCR master mix (Stratagene). Negative controls include a no-RT control
and no cDNA template control ($H_2O$ alone). Real-time PCR included initial denaturation at
95˚C for 3 min, followed by 40 cycles of 95˚C for 5 s, 60˚C for 20 sec, and 1 cycle of 95˚C for 1
min, 55˚C for 30 s, 95˚C for 30 s. The average of the collected data was normalized to the activ-
ity of a house-keeping gene, GAPDH. The relative expression (ΔΔCt) was calculated using
infected wild type HeLa cells or HEK-Blue TLR2 cells as a baseline.

## Co-immunoprecipitation and Western blots

*Immunoprecipitation* analysis of HeLa cells overexpressing MyD88-HA and with either
SIGIRR-FLAG or empty vector proteins was conducted using FLAG Immunoprecipitation Kit
(Sigma) following manufacturer's instructions. Briefly, LGV/L2 infected or uninfected HeLa
cells that overexpressed SIGIRR-FLAG protein were washed 3 times in PBS. Performing all
steps at 2–8˚C, the cells from each plate were lysed in 300 μl lysis buffer (Sigma), with protease
inhibitor cocktail, for 20 minutes on a shaker, and the lysates were collected by centrifuging
for 10 minutes at 12,000x g.

The cell lysates and positive controls were added to the thoroughly resuspended, washed
and pre-cleared ANTI-FLAG M2 resin (lysate:resin ratio is 20:1) and were left overnight on a
roller shaker. Samples were collected via several rounds of centrifugation (8000x g) and washes
in 1x wash buffer. To elute the samples, packed resins were boiled for 3 minutes in a 2x sample
buffer that did not contain any reducing agents. Samples were loaded on 12% SDS-PAGE gel
and immunoblotted using rabbit anti-SIGIRR (Abcam), mouse-anti-MyD88 (Millipore) or
mouse anti-FLAG M2 (Sigma) antibodies. The protein of interest was visualized using ECL
Plus Western blotting detection reagents (Millipore) following manufacturer's instructions.
Images of the bands and their intensity were detected and measured using a Gel Doc system
(Bio-Rad).

## Statistical analysis

The statistical significance was evaluated using GraphPad Instat software (GraphPad Software
Inc, La Jolla, CA) by unpaired Student's **t**-test. A value of $p < 0.05$ was considered significant.

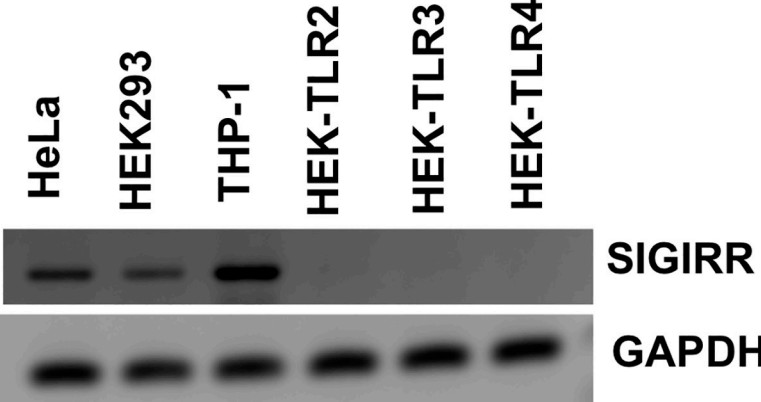

**Fig 1. Expression of SIGIRR in cell lines.** PCR analysis of SIGIRR in various human cell lines. Cervical epithelial HeLa cells, HEK293 and THP-1 cells express SIGIRR, while HEK-TLR2, HEK-TLR3 and HEK-TLR4 do not.

Data presented in each figure is the cumulative result of at least 3 experiments, unless stated otherwise.

## Results

### SIGIRR expression in different cell lines

The presence of SIGIRR was first screened by PCR in different cell lines that are commonly used for immunological studies. As shown in Fig 1, high levels of SIGIRR were detected in the human cervical carcinoma cell line HeLa and the human acute monocytic leukemia cell line THP-1, but slightly reduced levels were found in the human embryonic kidney cells HEK293. None of the modified HEK-Blue cells expressing human TLR2, TLR3 and TLR4 expressed detectable levels of SIGIRR mRNA.

### SIGIRR depletion results in higher levels of IL-8 mRNA during infection

To investigate the role of SIGIRR in *C. trachomatis*-infected cells, HeLa cells were first transfected with SIGIRR-specific siRNA. Real-time PCR analysis of the efficiency of the transfection indicated that there was no significant difference in expression levels of SIGIRR in HeLa cells treated with Lipofectamine only (Lipo Cont) or siRNA non-target control (siRNA CONT) compared with non-transformed (N.T.) cells (Fig 2A). However, both SIGIRR-siRNA specific sequences (sequence A or B) reduced mRNA levels by 40 and 80%, respectively, when compared with the non-transformed (N.T.) cells (Fig 2A).

The effect of SIGIRR on *C. trachomatis*-infected cells was tested by measuring cytokine expression using real-time PCR. After infection of HeLa cells with *C. trachomatis*, the expression levels of IL-8 were higher in cells that have reduced levels of SIGIRR compared with the non-transformed (N.T.) infected cells (Fig 2B). Both siRNA sequences used (SIGIRR-siRNA-A and B) increased the expression of IL-8. These data suggest that SIGIRR is involved in reducing the innate immune response against *C. trachomatis* infection.

### SIGIRR association with MyD88

Since MyD88 has been previously shown to localize on the *C. trachomatis* inclusion, where it interacts with TLR2 [32], we investigated whether SIGIRR can physically associate with MyD88 during *C. trachomatis* infection. HeLa cells transfected with 4.0 μg of MyD88-HA

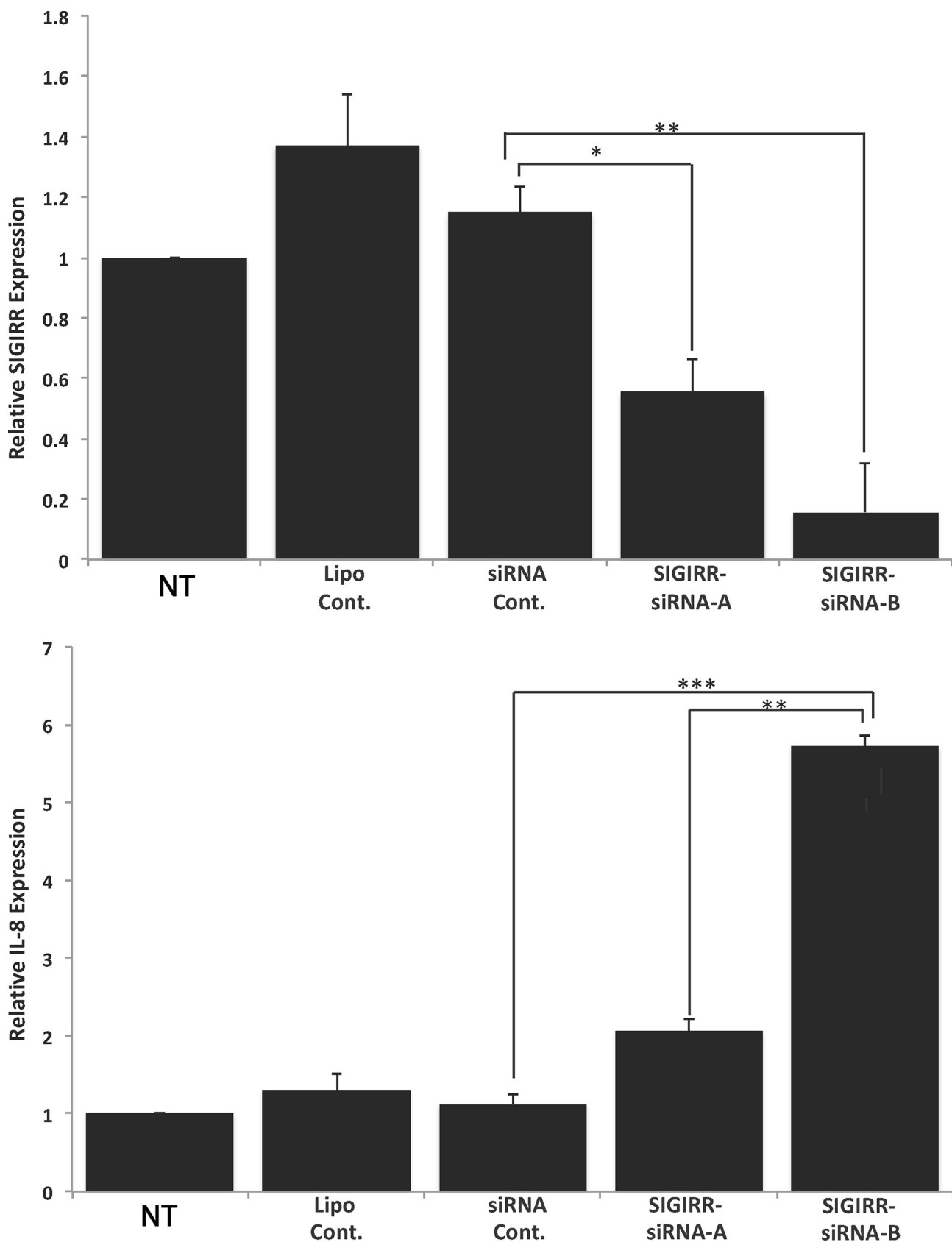

**Fig 2. Depletion of SIGIRR in HeLa cells induces higher levels of IL-8 mRNA during infection.** (A) HeLa cells were transfected with siRNA non-target control (siRNA CONT); SIGIRR-siRNA (sequence A or B); Lipofectamine treated (Lipo Cont) or non-transformed (N.T.). Relative expression level of SIGIRR compared with WT was reduced by 40 to 80% in SIGIRR-siRNA (panels A and B, respectively). (B) IL-8 expression in *C. trachomatis* serovar L2 infected HeLa cells, at MOI of 1 for 24 hours, showed higher levels of IL-8 expression in the cells that express lower levels of SIGIRR compared with WT infected cells. The increase of IL-8 expression inversely correlated with the expression levels of SIGIRR, as shown in panel A. Data were collected from 4 independent experiments. Error bars represent ±SD, and Student's t test was conducted. ns. (not significant), (*p < 0.05, **p < 0.01, ***p < 0.001).

along with 4.0 µg of SIGIRR-FLAG, or 4.0 µg of the pCDNA3.1+ empty vector, were either uninfected or infected with *C. trachomatis*. The co-immunoprecipitation results showed that MyD88 was associated with SIGIRR regardless of *C. trachomatis* infection (Fig 3, top panel, lanes 3 and 4). However, infected samples showed an increased intensity of the MyD88 band compared with uninfected samples, suggesting that *C. trachomatis* infection led to an increased association between MyD88 and SIGIRR.

## SIGIRR overexpression reduces the mRNA level of IL-8

Since our data from Fig 2B showed that SIGIRR deficiency induces the expression of IL-8, we next examined the impact of SIGIRR overexpression on *C. trachomatis* infected cells. HEK-Blue-hTLR2 cells were used for the overexpression experiments since they stably overexpress TLR2 and do not express endogenous SIGIRR (Fig 1A). HEK-Blue-hTLR2 cells were

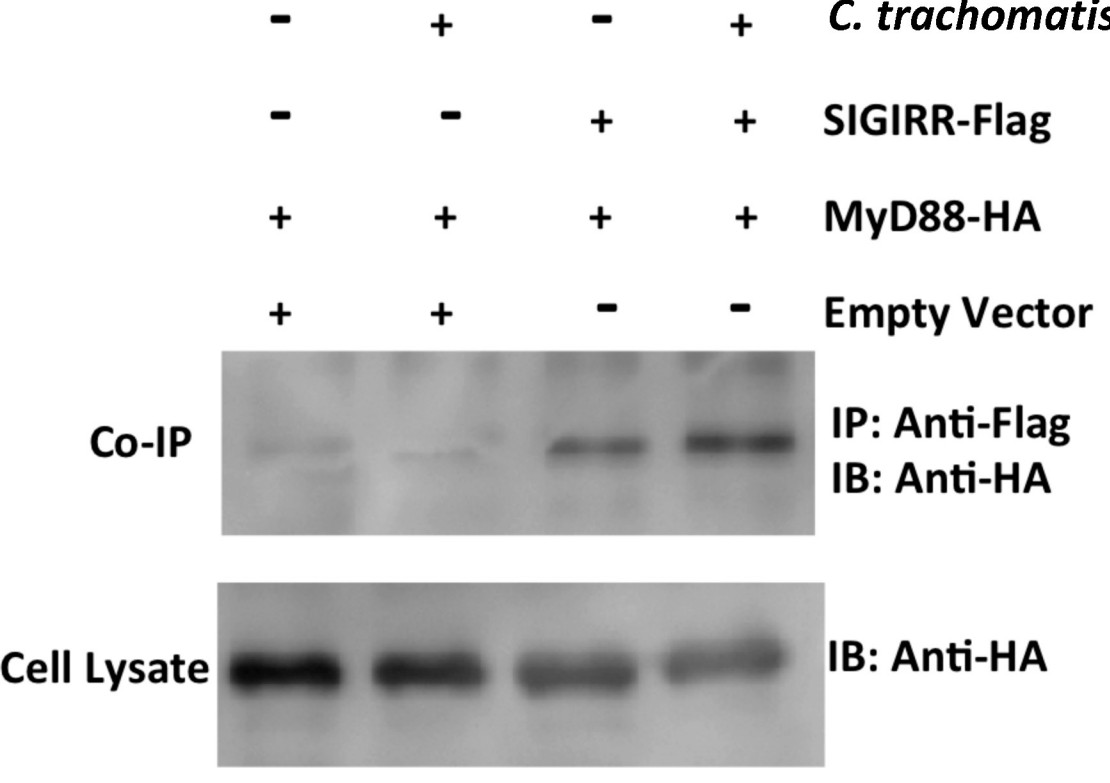

**Fig 3. SIGIRR associates with MyD88.** HeLa cells were transfected with 4.0 µg of MyD88-HA and either SIGIRR-FLAG construct or mock-transfected with 4.0 µg of pCDNA3.1+ empty vector. Forty-eight hours post transfection, cells were either left uninfected or infected with LGV/L2 as indicated. Seventy-two hours post transfection, cells were lysed and immunoprecipitated using FLAG Immunoprecipitation Kit (IP). Samples immunoblotted (IB) using anti-HA. Both L2 infected or uninfected HeLa cells showed an association of MyD88 with SIGIRR (top panel-right). Whole cell lysate indicated the presence of MyD88-HA in all the samples (bottom panel). Data shown is representative of 3 independent experiments.

transiently transfected or mock-transfected with 0.1 µg and 1.0 µg of SIGIRR-FLAG construct or pCDNA3.1+ empty vector, respectively. Real-time PCR showed that the level of SIGIRR mRNA in the transfected cells was proportional to the amount of plasmid DNA introduced into the cells, while cells transfected with the empty vector had, as expected, undetectable levels of SIGIRR (Fig 4A). Infecting the cells with *C. trachomatis* induced expression of IL-8 in both cells expressing SIGIRR-FLAG and empty vectors (Fig 4B). However, HEK-Blue-hTLR2 cells that expressed a low level of SIGIRR (0.1 µg) showed statistically significant, reduced levels of IL-8 compared with cells that did not overexpress SIGIRR (Fig 4B, left). On the other hand, cells transfected with a higher level of SIGIRR-FLAG also showed reduced levels of IL-8 mRNA, but the difference with cells that did not overexpress SIGIRR was not significant (Fig 4B, right).

## Discussion

Here we have shown that, in the absence of SIGIRR, human epithelial cells infected with *C. trachomatis* produce high levels of the pro-inflammatory cytokine IL-8. Also, overexpression analyses demonstrated that SIGIRR associates with MyD88 in infected and uninfected epithelial cells. These findings may help to better understand the mechanisms that lead *C. trachomatis* infections to induce significant pathology.

As a member of the Toll/IL-1 receptor superfamily that plays a role in activating the innate immune system, SIGIRR is expected to function as a regulator of these pathways. Indeed, studies have shown that SIGIRR regulates the immune response under various conditions [8, 33, 34]. Therefore, we hypothesized that SIGIRR may negatively regulate TLR signaling during *C. trachomatis* infection. SIGIRR expression in various human and murine tissues has been reported [11, 17]. Epithelial cells in particular express high levels of SIGIRR [17]. Our data shows that the human cell lines HeLa (cervical epithelium), HEK293 (human embryonic kidney cells) and THP1 (monocytic) all express SIGIRR, but not the HEK cells overexpressing TLR2, TLR3 or TLR4.

Production of proinflammatory cytokines is a crucial mechanism by which infected cells modulate innate and adaptive immune responses. For example, SIGIRR-deficient mice infected with *Mycobacterium tuberculosis* have higher mortality rates compared with the controls, in spite of no differences in tissue bacterial load in the lungs, liver or spleen. The increased mortality was thought to be due to an exuberant systemic inflammatory response, as demonstrated by enhanced macrophage and neutrophil lung infiltration and increased systemic levels of inflammatory cytokines. Importantly, in vivo depletion of mediators of the inflammatory responses (IL-1a and TNFα) in *M. tuberculosis* infected SIGIRR-deficient mice prolonged their survival [35]. The relevance of these findings is supported by a study that showed that three SNPs in the SIGIRR gene correlated with development of pulmonary and central nervous system tuberculosis in patients [36]. Other studies have reported contradictory data. For example, in mice with SIGIRR deficiency infected with *Streptococcus pneumoniae*, there was delayed mortality, reduction in the dissemination of the infection and reduced bacterial load in the lungs despite increased interstitial and perivascular inflammation in comparison to controls [37]. These findings suggest, as proposed by Molgora et al. [2], that during homeostasis, SIGIRR protects against unwanted responses, while its down-regulation during acute inflammatory stimulation enhances the antibacterial host defense, with a potential for pathogenic outcomes [2].

Since the IL-8 chemokine plays an important role during many bacterial infections, our study focused on the expression levels of IL-8 during *C. trachomatis* infection [38, 39]. In fact, we found that IL-8 levels increased in cells depleted of SIGIRR during *C. trachomatis* infection.

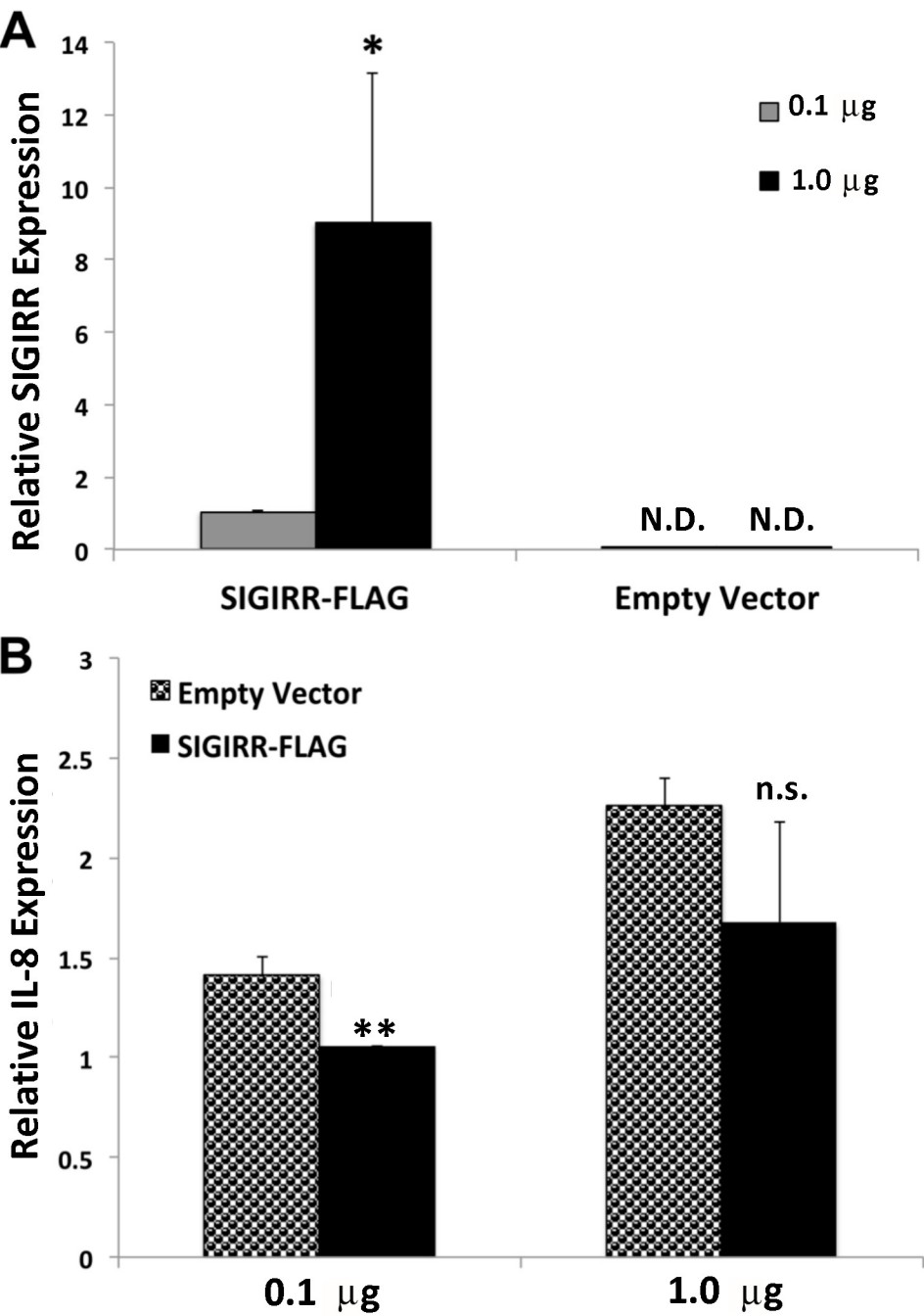

**Fig 4. Overexpression of SIGIRR slightly reduces expression of IL-8 mRNA after infection with *C. trachomatis*.** HEK-Blue-hTLR2 cells transiently transfected, or mock-transfected, with 0.1 μg or 1.0 μg of SIGIRR-FLAG construct or pCDNA3.1+ empty vector (A). The expression of SIGIRR in HEK-Blue hTLR2 cells was detected only in SIGIRR-FLAG transfected cells. The expression level of SIGIRR mRNA was proportional to the level of the plasmid DNA introduced into the cells. (B) Mock-cells and SIGIRR-FLAG infected with *C. trachomatis* at an MOI of 1 for 24 hrs. The mRNA expression levels of IL-8 were reduced in cells transfected with SIGIRR-FLAG construct compared with the empty vector. This reduction was statistically significant only at the lower concentration of SIGIRR (0.1 μg). Data were collected from 4 independent experiments. Error bars represent ±SD, and Student's t test was conducted. N.D. (not detected), n.s. (not significant), $^*p < 0.05$, $^{**}p < 0.01$.

A report by O'Connell *et al.* showed that TLR2 and MyD88 co-localize around the *C. trachomatis* inclusion of infected cells [40]. Our co-immunoprecipitation studies showed that there is a specific association between SIGIRR and MyD88 in both infected and non-infected HeLa cells. However, there is an increased association between SIGIRR and MyD88 during infection with *C. trachomatis* infection, suggesting that SIGIRR may also associate with the chlamydial inclusion.

Since depleting SIGIRR resulted in higher levels of IL-8 expression during infection, we expected that SIGIRR overexpression would result in lower levels of IL-8. In fact, the levels of IL-8 were lower in cells that overexpressed SIGIRR, compared with mock-transfected cells.

In conclusion, we have shown that depletion of SIGIRR results in increased levels of IL-8 in *C. trachomatis*-infected cells. Our data suggest that SIGIRR localizes in the membrane and the cytosol and associates with MyD88 in uninfected cells, but some SIGIRR may also associate with the *C. trachomatis* inclusion during infection, according to our immunoprecipitation results. Altogether, our results suggest that SIGIRR is involved in signal transduction via TLRs during chlamydial infection, and down-regulates this signaling pathway. We predict that polymorphisms in the human SIGIRR gene that increase or decrease activity of the SIGIRR protein will also affect the intensity of inflammation and pathogenesis resulting from a *C. trachomatis* infection.

## Supporting information

**S1 Raw images.**
(PDF)

## Author Contributions

**Data curation:** Mufadhal Al-Kuhlani.

**Formal analysis:** Mufadhal Al-Kuhlani, Luis de la Maza, David M. Ojcius.

**Funding acquisition:** Luis de la Maza, David M. Ojcius.

**Investigation:** Mufadhal Al-Kuhlani, Graham Lambert, Sukumar Pal.

**Project administration:** David M. Ojcius.

**Supervision:** Luis de la Maza, David M. Ojcius.

**Writing – original draft:** Mufadhal Al-Kuhlani, David M. Ojcius.

**Writing – review & editing:** Luis de la Maza, David M. Ojcius.

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
