## [Decision Letter · Decision Letter 0]

21 Jan 2020

PONE-D-19-34581

Immune Response Against Chlamydia trachomatis via Toll-Like Receptors Is Negatively Regulated by SIGIRR

PLOS ONE

Dear Dr. Ojcius,

Thank you for submitting your manuscript to PLOS ONE. After careful consideration, we feel that it has merit but does not fully meet PLOS ONE’s publication criteria as it currently stands. Therefore, we invite you to submit a revised version of the manuscript that addresses the points raised during the review process.

Reviewer #1 found your study interesting but also raised some concerns: While it may not be necessary to re-run the samples using qRT-PCR for Fig1, please explain why gel separation was used for Fig. 1 while qRT-PCR for Fig.2. It is also important for you to correct and discuss other issues raised by the reviewer. Finally, for Fig. 3, the reviewer requested quantitative data. I agree that it will strengthen the manuscript if you can ask your students to provide semi-quantitative data.    

We would appreciate receiving your revised manuscript by Mar 06 2020 11:59PM. To enhance the reproducibility of your results, we recommend that if applicable you deposit your laboratory protocols in protocols.io, where a protocol can be assigned its own identifier (DOI) such that it can be cited independently in the future. For instructions see: http://journals.plos.org/plosone/s/submission-guidelines#loc-laboratory-protocols

We look forward to receiving your revised manuscript.

Kind regards,

Guangming Zhong

Academic Editor

PLOS ONE

Journal Requirements:

Reviewers' comments:

Reviewer's Responses to Questions

**Comments to the Author**

1. Is the manuscript technically sound, and do the data support the conclusions?

Reviewer #1: Partly

2. Has the statistical analysis been performed appropriately and rigorously? 

Reviewer #1: Yes

3. Have the authors made all data underlying the findings in their manuscript fully available?

Reviewer #1: No

4. Is the manuscript presented in an intelligible fashion and written in standard English?

Reviewer #1: Yes

5. Review Comments to the Author

Reviewer #1: Previous studies have shown that Chlamydia trachomatis infection activates TLR-mediated signaling pathways. Now, Al-Kuhlani et al demonstrated a potential inhibitory role for SIGIRR in TLR signaling in C. trachomatis-infected epithelial cells. Whereas their findings are interesting, the manuscript also has some weaknesses:

1. The expression analyses of SIGIRR in Fig. 1 were done with reverse transcription PCR. The PCR products were analyzed using gel electrophoresis. This is an antiquated technique, and these kinds of results are no longer acceptable to many journals. Given the fact they used to reverse transcription quantitative PCR to generate Fig. 2, they should reanalyze their samples collected for Fig. 1 with quantitative PCR.

2. Given the fact that levels of mRNA and protein are not always in agreement, it would be nice if they perform protein level analysis if antibodies are available. If protein analysis is not feasible, it is important that the authors state mRNA was analyzed in the subheadings of Results and figure titles. The potential discordance between mRNA and protein levels should be stated in the Discussion.

3. For both main text leading to Fig. 1 and the legend, the detection technique should be reverse transcription PCR instead of just PCR.

4. The rationale for selecting the cell lines for SIGIRR expression level analysis in Fig. 1 should be stated. Assuming that it is true that SIGIRR levels in HEK cells are reduced by TLR2, 3, 4 over expression, they should discuss potential implications of the findings and/or underlying mechanisms.

5. In the context of Fig. 2, it is probably more appropriate to change WT to something like NT (non-transformed).

6. Fig. 3 is representative of 3 experiments. They should present quantitative data like in other figures in addition to images.

6. PLOS authors have the option to publish the peer review history of their article (what does this mean?). If published, this will include your full peer review and any attached files.

Reviewer #1: No

---

## [Author Response · Author response to Decision Letter 0]

20 Feb 2020

Reviewer #1: Previous studies have shown that Chlamydia trachomatis infection activates TLR-mediated signaling pathways. Now, Al-Kuhlani et al demonstrated a potential inhibitory role for SIGIRR in TLR signaling in C. trachomatis-infected epithelial cells. Whereas their findings are interesting, the manuscript also has some weaknesses:

1. The expression analyses of SIGIRR in Fig. 1 were done with reverse transcription PCR. The PCR products were analyzed using gel electrophoresis. This is an antiquated technique, and these kinds of results are no longer acceptable to many journals. Given the fact they used to reverse transcription quantitative PCR to generate Fig. 2, they should reanalyze their samples collected for Fig. 1 with quantitative PCR.

Authors’ response: The goal for the gel separation was to check for the presence or absence of SIGIRR in these cell lines before we proceeded to introduce the siRNA or vector. Our goal was not to quantify the expression, but simply to determine if these cells were good candidates for the planned experiments (specially the HEK-293-TLRs cells, as they had never been studied before with regards to SIGIRR). Given the large difference between the cell lines, which could be observed even visually, we did not pursue further the quantification of SIGIRR expression.

2. Given the fact that levels of mRNA and protein are not always in agreement, it would be nice if they perform protein level analysis if antibodies are available. If protein analysis is not feasible, it is important that the authors state mRNA was analyzed in the subheadings of Results and figure titles. The potential discordance between mRNA and protein levels should be stated in the Discussion.

Authors’ response: In fact, antibodies against SIGIRR are not available. Following the referee’s advice, we therefore state that mRNA was analyzed in the Results and figure legends. 

3. For both main text leading to Fig. 1 and the legend, the detection technique should be reverse transcription PCR instead of just PCR.

Authors’ response: As mentioned in the response to point 1, the difference in mRNA expression was so dramatically different between the different cell lines (which was clearly obvious to the naked eye), we did not pursue further quantification of SIGIRR expression.

4. The rationale for selecting the cell lines for SIGIRR expression level analysis in Fig. 1 should be stated. Assuming that it is true that SIGIRR levels in HEK cells are reduced by TLR2, 3, 4 over expression, they should discuss potential implications of the findings and/or underlying mechanisms.

Authors’ response: Hela cells are the cell of choice for studies of Chlamydia infection, and we therefore verified first that SIGIRR is expressed in HeLa cells. HEK-293 do not express SIGIRR and TLRs, but the stably-transfected HEK cells (HEK-Blue with TLRs) provide a convenient way of measuring NF-�B activation under different conditions. THP-1 served as control for SIGIRR expression since it is known that they express high levels of the protein [Kadota et al, Clin Exp Immunol 2010; 162:348-61].

The HEK cells and the stably-transfected HEK-TLR cells are commercially available. Like the referee, we were intrigued by the observation that the HEK-TLR cells do not express significant amounts of SIGIRR. However, we did not prepare these cells and, unfortunately, do not have an explanation for this result.

5. In the context of Fig. 2, it is probably more appropriate to change WT to something like NT (non-transformed).

Authors’ response: We are grateful to the referee for this advice. We therefore made the change in the figure and the text, as recommended.

6. Fig. 3 is representative of 3 experiments. They should present quantitative data like in other figures in addition to images.

Authors’ response: Similar to our analysis of Figure 1, the results of the PCR were dramatically different for the different conditions (which could be observed easily by the naked eye). Therefore, we did not pursue further the quantification of this experiment.

---

## [Editor Report · Decision Letter 1]

9 Mar 2020

Immune Response Against Chlamydia trachomatis via Toll-Like Receptors Is Negatively Regulated by SIGIRR

PONE-D-19-34581R1

Dear Dr. Ojcius,

We are pleased to inform you that your manuscript has been judged scientifically suitable for publication and will be formally accepted for publication once it complies with all outstanding technical requirements.

With kind regards,

Guangming Zhong

Academic Editor

PLOS ONE

Additional Editor Comments (optional):

The authors have adequately addressed all concerns raised by the reviewers.
---

## [Editor Report · Acceptance letter]

11 Mar 2020

PONE-D-19-34581R1 

Immune Response Against *Chlamydia trachomatis* via Toll-Like Receptors Is Negatively Regulated by SIGIRR 

Dear Dr. Ojcius:

I am pleased to inform you that your manuscript has been deemed suitable for publication in PLOS ONE. Congratulations! Your manuscript is now with our production department. 

With kind regards,

on behalf of

Dr. Guangming Zhong 

Academic Editor

PLOS ONE